# Neural signatures of arbitration between Pavlovian and instrumental action selection

**Samuel J. Gershman** [1,2]*, **Marc Guitart-Masip** [3,4], **James F. Cavanagh** [5]

**1** Department of Psychology and Center for Brain Science, Harvard University, Cambridge, Massachusetts, United States of America, **2** Center for Brains, Minds and Machines, MIT, Cambridge, Massachusetts, United States of America, **3** Max Planck-UCL Centre for Computational Psychiatry and Ageing Research, London, United Kingdom, **4** Aging Research Center, Karolinska Institute, Solna, Sweden, **5** Department of Psychology, University of New Mexico, Albuquerque, New Mexico, United States of America

* gershman@fas.harvard.edu

**Data Availability Statement:** All code and data for reproducing the analyses and figures is available at https://github.com/sjgershm/GoNoGo-neural.

## Abstract

Pavlovian associations drive approach towards reward-predictive cues, and avoidance of punishment-predictive cues. These associations "misbehave" when they conflict with correct instrumental behavior. This raises the question of how Pavlovian and instrumental influences on behavior are arbitrated. We test a computational theory according to which Pavlovian influence will be stronger when inferred controllability of outcomes is low. Using a model-based analysis of a Go/NoGo task with human subjects, we show that theta-band oscillatory power in frontal cortex tracks inferred controllability, and that these inferences predict Pavlovian action biases. Functional MRI data revealed an inferior frontal gyrus correlate of action probability and a ventromedial prefrontal correlate of outcome valence, both of which were modulated by inferred controllability.

## Author summary

Using a combination of computational modeling, neuroimaging (both EEG and fMRI), and behavioral analysis, we present evidence for a dual-process architecture in which Pavlovian and instrumental action values are adaptively combined through a Bayesian arbitration mechanism. Building on prior research, we find neural signatures of this arbitration mechanism in frontal cortex. In particular, we show that trial-by-trial changes in Pavlovian influences on action can be predicted by our computational model, and are reflected in midfrontal theta power, as well as inferior frontal and ventromedial prefrontal cortex fMRI responses.

## Introduction

Approaching reward-predictive stimuli and avoiding punishment-predictive stimuli are useful heuristics adopted by many animal species. However, these heuristics can sometimes lead animals astray—a phenomenon known as "Pavlovian misbehavior" [1, 2]. For example, reward-predictive stimuli invigorate approach behavior even when such behavior triggers

**Funding:** SJG was supported by the Center for Brains, Minds and Machines (CBMM), funded by NSF STC award CCF-1231216 (https://nsf.gov/), and by the Office of Naval Research (N00014-17-1-2984; https://www.onr.navy.mil/). MGM was supported by a research grant awarded by the Swedish Research Council (VR-2018-02606; https://www.vr.se/english.html). JFC was supported by NIMH 1RO1MH119382-01(url https://www.nimh.nih.gov/index.shtml). The funders had no role in study design, data collection and analysis, decision to publish, or preparation of the manuscript.

**Competing interests:** The authors have declared that no competing interests exist.

withdrawal of the reward [3, 4], or the delivery of punishment [5, 6]. Likewise, punishment-predictive stimuli inhibit approach behavior even when doing so results in reduced net reward [7–9].

A venerable interpretation of these and related findings is that they arise from the interaction between Pavlovian and instrumental learning processes [10]. The two-process interpretation has been bolstered by evidence from neuroscience that Pavlovian and instrumental influences on behavior are (at least to some extent) segregated anatomically [11]. In particular, the dorsal subdivision of the striatum (caudate and putamen in primates) is more closely associated with instrumental learning, whereas the ventral subdivision (nucleus accumbens) is more closely associated with Pavlovian learning [12, 13].

Any multi-process account of behavior naturally raises the question of arbitration: what decides the allocation of behavioral control to particular processes at any given point in time? One way to approach this question from a normative perspective is to analyze the computational trade-offs realized by different processes. The job of the arbitrator is to determine which process achieves the optimal trade-off for a particular situation. This approach has proven successful in understanding arbitration between different instrumental learning processes [14–17]. More recently, it has been used to understand arbitration between Pavlovian and instrumental processes [18]. The key idea is that instrumental learning is more statistically flexible, in the sense that it can learn reward predictions that are both action-specific and stimulus-specific, whereas Pavlovian learning can only learn stimulus-specific predictions. The cost of this flexibility is that instrumental learning is more prone to *over-fitting*: for any finite amount of data, there is some probability that the learned predictions will generalize incorrectly in the future, and this probability is larger for more flexible models, since they have more degrees of freedom with which to capture noise in the data. This account can be formalized in terms of Bayesian model comparison [18].

Dorfman and Gershman [18] tested the controllability prediction more directly using a variant of the Go/NoGo paradigm, which has been widely employed as an assay of Pavlovian bias in human subjects [19–24]. We focus on this task in the present paper, while acknowledging that our conclusions may not generalize to other forms of Pavlovian bias, such as in Pavlovian-instrumental transfer paradigms. The Go/NoGo task crosses valence (winning reward vs. avoiding punishment) with action (Go vs. NoGo), resulting in four conditions: Go-to-Win, Go-to-Avoid, NoGo-to-Win, and NoGo-to-Avoid (Fig 1A). A key finding from this paradigm is that people make more errors on Go-to-Avoid trials compared to Go-to-Win trials, and this pattern reverses for NoGo trials, indicating that Pavlovian bias invigorates approach (the Go response) for reward-predictive cues, and inhibits approach for punishment-predictive cues. By introducing decoy trials in which rewards were either controllable or uncontrollable, Dorfman and Gershman showed that the Pavlovian bias was enhanced in the low controllability condition (see also [25]).

An important innovation of the Dorfman and Gershman model was the hypothesis that the balance between Pavlovian and instrumental influences on action is dynamically arbitrated, and hence can potentially vary within the course of a single experimental session. This contrasts with most modeling of the Go/NoGo task (starting with [22]), which has assumed that the balance is fixed across the experimental session. Dorfman and Gershman presented behavioral evidence for within-session variation of the Pavlovian bias. Neural data could potentially provide even more direct evidence, by revealing correlates of the arbitration process itself. We pursue this question here by carrying out a model-based analysis of two prior data sets, one from an electroencephalography (EEG) study [23], and one from a functional magnetic resonance imaging (fMRI) study [22].

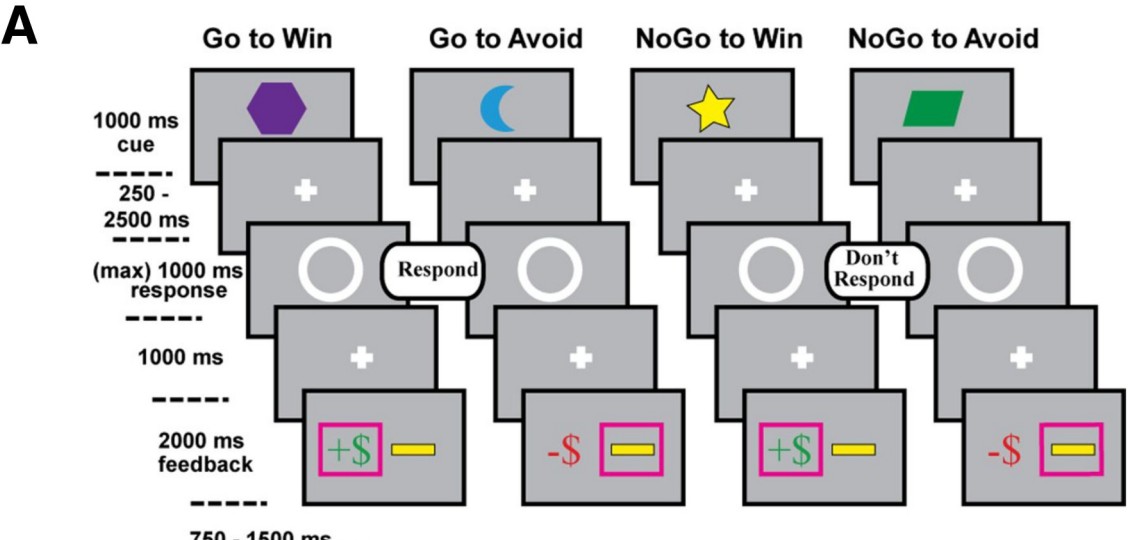

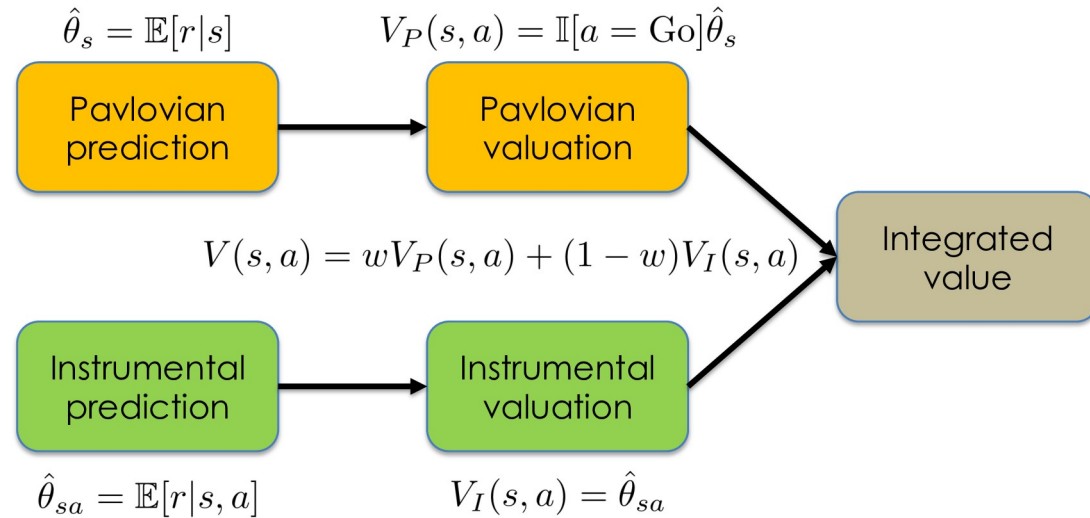

**Fig 1. Experimental design and computational framework.** (A) Shown here is the experimental design used by [23] in their EEG study, which differed in several minor ways from the design used by [22] in their fMRI study (see Materials and methods). Subjects were instructed to respond to a target stimulus (white circle) by either pressing a button (Go) or witholding a button (NoGo). Subjects had to learn the optimal action based on stimulus cues (shapes) and reward or punishment feedback. For all conditions, the optimal action yielded reward delivery or punishment avoidance with 70% probability; this probability was 30% for the suboptimal action. (B) Pavlovian and instrumental prediction and valuation combine into a single integrated decision value based on a weighting parameter (*w*) that represents the evidence for the uncontrollable environment (i.e., in favor of the Pavlovian predictor). Figure adapted from [18], with permission. See Materials and methods for technical details.

## Results

### Modeling and behavioral results

We fit computational models to Go/NoGo data from two previously published studies. The tasks used in these two studies were very similar, with a few minor differences detailed in the

Materials and Methods. We will first briefly summarize the models (more details can be found in the Materials and methods).

In [18], a Bayesian framework was introduced that formalized action valuation in terms of probabilistic inference (Fig 1B). According to this framework, Pavlovian and instrumental processes correspond to distinct predictive models of reward (or punishment) outcomes. The Pavlovian process estimates outcome predictions based on stimulus information alone, whereas the instrumental process uses both stimulus and action information. These predictions are converted into action values in different ways. For the instrumental process, action valuation is straightforward—it is simply the expected outcome for a particular stimulus-action pair. The Pavlovian process, which does not have an action-dependent outcome expectation, instead relies on the heuristic that reward-predictive cues should elicit behavioral approach (Go actions in the Go/NoGo task), and punishment-predictive cues should elicit avoidance (NoGo).

Arbitration in the Bayesian framework corresponds to model comparison: the action values are weighted by the probability favoring each predictor. This computation yields the expected action value under model uncertainty. Thus, the Bayesian framework offers an interpretation of Pavlovian bias in terms of the probability favoring Pavlovian outcome prediction (denoted by $w$, which we refer to as the "Pavlovian weight"). The Pavlovian weight can also be interpreted as the subjective degree of belief in an uncontrollable environment, where actions do not influence the probability distribution over outcomes (and correspondingly, $1 - w$ is the degree of belief in a controllable environment).

Dorfman and Gershman [18] compared two versions of probabilistic arbitration. In the Fixed Bayesian model, the Pavlovian weight reflects *a priori* beliefs (i.e., prior to observing data). Thus, in the Fixed Bayesian model, the Pavlovian bias weight does not change with experience. In the Adaptive Bayesian model, the Pavlovian weight reflects *a posteriori* beliefs (i.e., after observing data), such that the weight changes across trials based on the observations. Finally, we compared both Bayesian models to a non-Bayesian reinforcement learning (RL) model that best described the data in [22]. This RL model is structurally similar to the Fixed Bayesian model, but posits a heuristic aggregation of Pavlovian and instrumental values. All models use an error-driven learning mechanism, but the Bayesian models assume that the learning rate decreases across stimulus repetitions.

We found that the Adaptive Bayesian model was favored in both data sets, with a protected exceedance probability greater than 0.7 (Fig 2A and 2B). To confirm that the Adaptive model fit the data well (see S1 Fig for further predictive checks), we plotted the go bias (difference in accuracy between Go and NoGo trials) as a function of weight quantile (Fig 2C and 2D). Consistent with the model and previous results [18], the go bias increased with weight for the Win condition [top vs. bottom quantile: $t(31) = 2.41$, $p < 0.05$ for the EEG data set, $t(28) = 3.96$, $p < 0.001$ for the fMRI data set]. In contrast, it remained essentially flat for the Avoid condition. This asymmetry arises from the fact that most subjects were best fit with an initial Pavlovian value greater than 0 (76% in the EEG data set, 63% in the fMRI data set). This means that the model actually predicts a *positive* go bias for the Avoid condition early during learning (when the Pavlovian weight is typically larger; see S2 Fig), which eventually should become a negative go bias. Consistent with this hypothesis, the go bias during the first 40 trials (across all conditions) in the fMRI data set was significantly greater than 0 for the Avoid condition [$t(29) = 3.63$, $p < 0.002$] and significantly less than 0 during the last 40 trials [$t(29) = 2.24$, $p < 0.05$] (the EEG data set had fewer trials, and hence it was harder to obtain a reliable test of this hypothesis, though the results were numerically in the same direction). This early preference for the go response is also consistent with earlier modeling of these data sets in which an

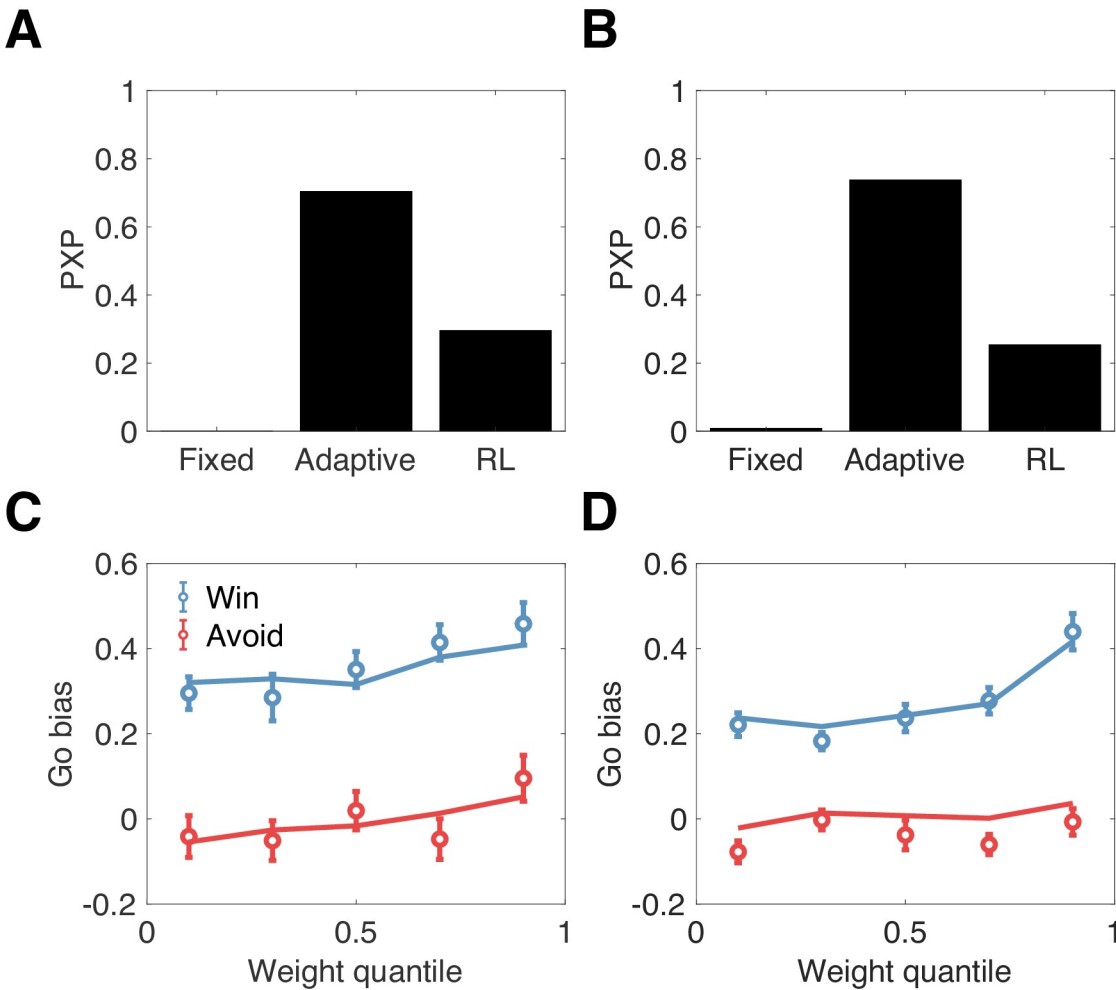

**Fig 2. Behavioral results.** Top: Protected exceedance probabilities (PXPs) for 3 computational models fit to the EEG data set (A) and the fMRI data set (B). Bottom: Go bias (difference in accuracy between Go and NoGo trials) computed as a function of the Pavlovian weight for the EEG data set (C) and the fMRI data set (D). Lines show model fits, circles show means with standard errors.

unlearned go bias was incorporated into the decision value (see also [9]). Here we explain the same phenomenon in terms of the prior over Pavlovian values.

In the next two sub-sections, we use the Adaptive model to generate model-based regressors for neural activity, in an effort to ground the hypothesized computational processes. In particular, we will focus on showing that neural signals covary with the Pavlovian weight, thereby demonstrating that this dynamically changing variable is encoded by the brain. Before proceeding to these analyses, it is important to show that this covariation is not confounded by other dynamic variables. In particular, while the Fixed and RL models lack a dynamic weight, the instrumental and Pavlovian values are dynamic in these models. To eliminate these variables as potential confounds, we correlated them with the Pavlovian weight for each subject. For both the EEG and fMRI data sets, the median correlation never exceeded 0.02, and the median correlation never significantly differed from 0 ($p > 0.1$, signed rank test). To evaluate the positive evidence for the null hypothesis (correlation of 0), we also carried out Bayesian t-tests [26] on the Fisher z-transformed correlations, finding that the Bayes factors consistently favored the null hypothesis (ranging between 2 and 5). These results gives us confidence that

**A**                                              **B**

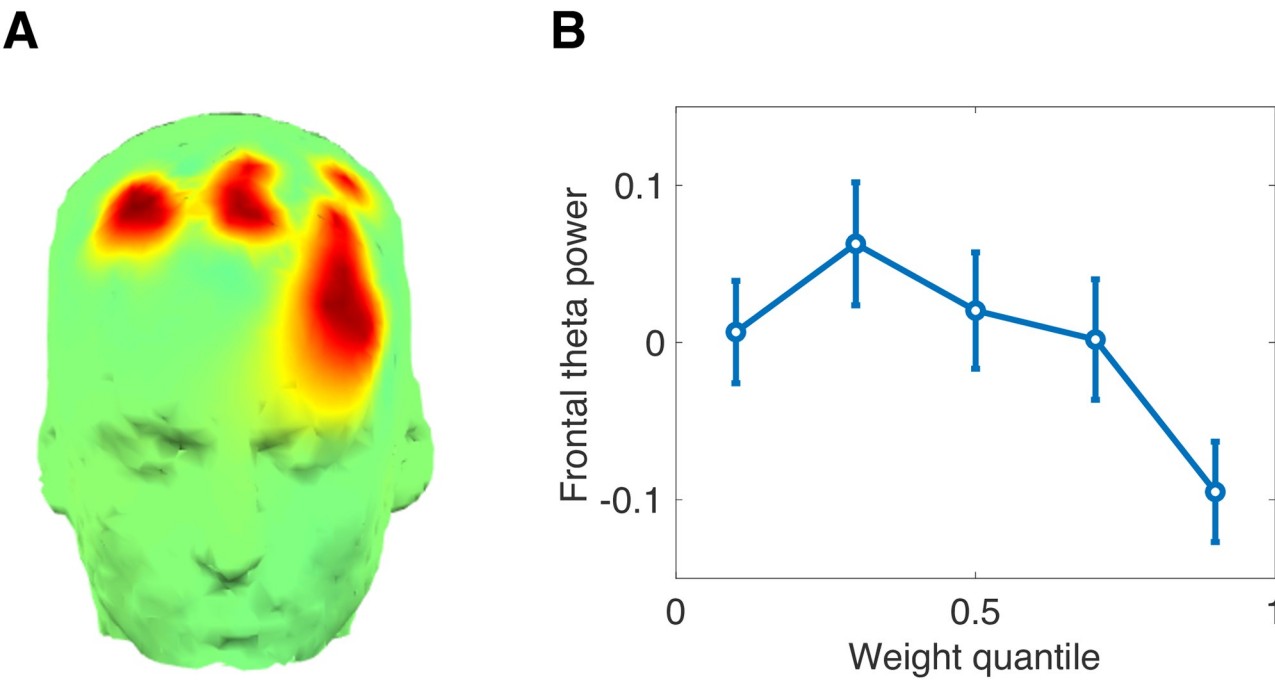

**Fig 3. EEG results.** (A) Montage showing region of interest, derived from [23]. (B) Stimulus-locked midfrontal theta power as a function of the Pavlovian weight. Error bars show standard error of the mean.

the neural covariation we report next is unconfounded by other dynamic variables in the computational model.

## EEG results

Following the template of our behavioral analyses, we examined midfrontal theta power as a function of the Pavlovian weight (see S3 Fig for results fully disaggregated across conditions). In previous work on this same data set [23], and in follow-up studies [25, 27], frontal theta was implicated in the suppression of the Pavlovian influence on choice. Consistent with these previous findings, we found that frontal theta power decreased with the Pavlovian weight [top vs. bottom quantile: $t(31) = 2.09$, $p < 0.05$; Fig 3]. Unlike these earlier studies, which incorporated the frontal theta signal as an input into the computational model, we have validated for the first time a model of the frontal theta signal (i.e., as an output of the model).

Our data cannot be explained by alternative theories about midfrontal theta. First, we can rule out an explanation in terms of reward prediction error [28–30]. At the time of stimulus presentation, the reward prediction error is simply the estimate of stimulus value (a recency-weighted average of past prediction errors), which is uncorrelated with $w$ ($p = 0.29$; signed rank test; Bayes factor favoring the null: 2.36, Bayesian t-test applied to Fisher z-transformed correlations) as well as with midfrontal theta ($p = 0.23$, signed rank test; Bayes factor favoring the null: 2.62). Second, our data are not adequately explained by choice confidence [31]. Although we do not measure subjective choice confidence, we can use as a proxy the expected accuracy under our model. This measure is negatively correlated with the Pavlovian weight ($p < 0.001$, signed rank test), as we would expect given that accuracy will tend to be lower under Pavlovian control. However, repeating our quantile comparison using this measure did

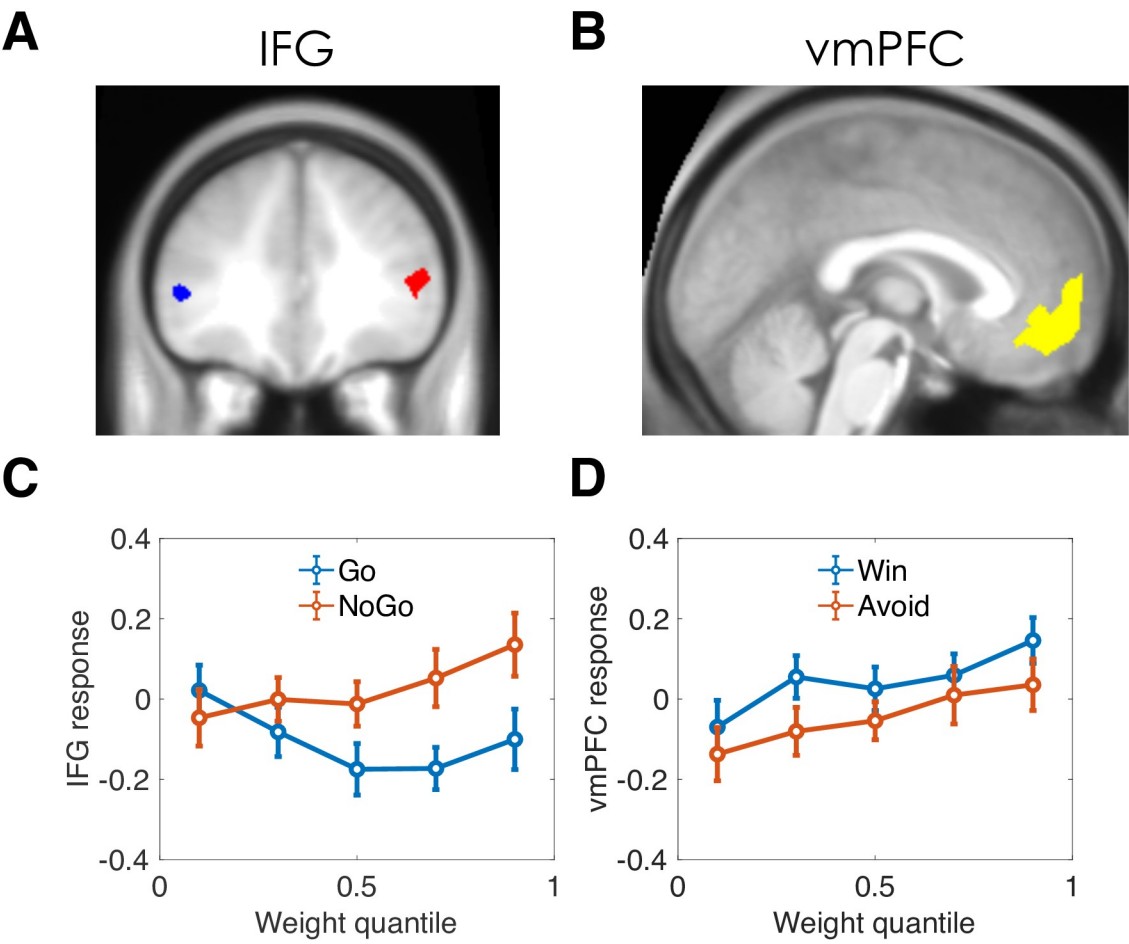

**Fig 4. Functional MRI results.** (A,B) Regions of interest. IFG: inferior frontal gyrus; vmPFC: ventromedial prefrontal cortex. (C) IFG response to the stimulus cue as a function of Pavlovian weight, separated by Go and NoGo conditions. (D) vmPFC response to the stimulus cue as a function of Pavlovian weight, separated by Win and Avoid conditions.

not reveal a significant relationship between expected accuracy and midfrontal theta power (top vs. bottom quantile: $p = 0.14$).

## fMRI results

We next re-analyzed fMRI data from [22], focusing on two frontal regions of interest: the inferior frontal gyrus (IFG; Fig 4A) and the ventromedial prefrontal cortex (vmPFC; Fig 4B). The key results are summarized in panels C and D in Fig 4 (see S4 Fig for fully disaggregated results, including results from the ventral striatum). When the Pavlovian weight is close to 0, the IFG response to the cue for Go and NoGo conditions is not significantly different ($p = 0.32$), but when the Pavlovian weight is close to 1, IFG responds significantly more to NoGo than to Go [$t(28) = 3.91$, $p < 0.001$, Fig 4C]. This is consistent with the hypothesis that IFG is responsible for the suppression of Go responses when the Pavlovian bias is strong, regardless of valence. Note that the NoGo>Go effect is unsurprising given that the IFG region of interest was selected based on the NoGo>Go contrast, but this selection criterion does not by itself explain the interaction between weight and NoGo vs. Go.

The vmPFC scaled with the Pavlovian weight [top vs. bottom quantile: $t(28) = 2.05$, $p < 0.05$; Fig 4D], and responded more to Win vs. Avoid across weight quantiles [$t(28) = 3.46$, $p < 0.002$], but the interaction was not significant ($p = 0.56$). Thus, vmPFC appears to encode a combination of valence and Pavlovian bias (both main effects but no interaction). The relationship between Pavlovian weight and vmPFC cannot be explained by choice confidence (again using expected accuracy as a proxy): although we expect expected accuracy to vary inversely with the Pavlovian weight (see above), vmPFC activity did not differ significantly between the top and bottom quantile of expected accuracy ($p = 0.08$). Nor can the relationship be explained by differences in instrumental values, which are uncorrelated with the Pavlovian weight (median $r = 0.02$, $p = 0.36$, signed rank test; Bayes factor favoring the null: 3.30, Bayesian t-test applied to Fisher z-transformed correlations).

## Discussion

By re-analyzing two existing neuroimaging data sets, we have provided some of the first evidence for neural signals tracking beliefs about controllability during Go/NoGo task performance. These signals are theoretically significant, as they support the computational hypothesis that Pavlovian influences on choice behavior arise from a form of Bayesian model comparison between Pavlovian and instrumental outcome predictions [18]. Modeling of behavior further supported this hypothesis, showing that the behavioral data were best explained by a Bayesian model in which Pavlovian influence changes as a function of inferred controllability.

Our analyses focused on three regions, based on prior research. One strong point of our approach is that we did not select the regions of interest based on any of the analyses reported in this paper; thus, the results serve as relatively unbiased tests of our computational hypotheses.

First, we showed that midfrontal theta power tracked inferred controllability (i.e., inversely with the Pavlovian weight). This finding is consistent with the original report describing the data set [23], which showed that the Pavlovian weight governing action selection could be partially predicted from midfrontal theta power, a finding further supported by subsequent research [27]. A recent study [25] attempted to more directly link midfrontal theta to controllability using a "learned helplessness" design in which one group of subjects intermittently lost control over outcomes by "yoking" the outcomes to those observed by a control group. The control group exhibited the same relationship between Pavlovian weight and midfrontal theta observed in earlier studies, whereas the yoked group did not (however, it must be noted that a direct comparison did not yield strong evidence for group differences). More broadly, these results are consistent with the hypothesis that midfrontal theta (and its putative cortical generator in midcingulate / dorsal anterior cingulate cortex) is responsible for computing the "need for control" [32, 33] or the "expected value of control" [34]. Controllability is a necessary (though not sufficient) requirement for the exertion of cognitive control to have positive value. From this perspective, it is important to emphasize that we do not see midfrontal theta as exclusively signaling inferred controllability; rather, high inferred controllability signals need for control, which thereby evokes midfrontal theta activity. Other variables that signal need for control, such as reward prediction error [28–30], can also evoke midfrontal theta activity, without requiring changes in inferred controllability.

Because of the partial volume acquisition in the imaging procedure (see Materials and methods), our fMRI data did not allow us to examine hemodynamic correlates in the midcingulate cortex. Instead, we examined two other regions of interest: IFG and vmPFC. IFG has been consistently linked to inhibition of prepotent responses [35, 36]. Accordingly, we found

greater response to NoGo than to Go in IFG. However, this difference only emerged when inferred controllability (as determined by our computational model) was low. There is some previous evidence that IFG is sensitive to controllability. Romaniuk and colleagues [37] reported that the IFG response was stronger on free choice trials compared to forced choice trials, and was significantly correlated with self-reported ratings of personal autonomy. Similarly, IFG activity has been associated with illusions of control [38]. It is difficult to directly connect these previous findings with those reported here, since the studies did not compare Go and NoGo responses.

While our finding that vmPFC shows a stronger response to reward vs. punishment is consistent with previous findings [39, 40], the fact that vmPFC decreases with inferred controllability is rather surprising. If anything, the literature suggests that vmPFC *increases* with subjective and objective controllability [41–43], though at least one study found a greater *reduction* in vmPFC activity after a controllable punishment compared to an uncontrollable punishment [44]. Further investigation is needed to confirm the surprising inverse relationship between vmPFC and inferred controllability.

Our study is limited in a number of ways, which point toward promising directions for future research. First, as already mentioned, our fMRI data did not allow us to test the hypothesis that midcingulate cortex, as the putative generator of midfrontal theta, tracked inferred controllability. This limitation could be overcome in future studies using whole brain acquisition volumes. Second, we only analyzed neural data time-locked to the stimulus; future work could examine outcome-related activity. We chose not to do this because of our focus on action selection, and in particular how inferred controllability signals are related to Pavlovian biasing of actions. An important task for future work will be to identify the neural update signal for inferred controllability that drives dynamic changes in Pavlovian bias. Third, while our work is partly motivated by studies of Pavlovian-instrumental transfer [45], it is still an open empirical question to what extent that phenomenon is related to Pavlovian biases in Go/NoGo performance. It is also an open theoretical question to what extent the kind of Bayesian arbitration model proposed here can provide a comprehensive account of Pavlovian-instrumental transfer. Finally, in accordance with most prior analyses of midfrontal theta, we focused only on total theta power, neglecting any distinctions between phase-locked and non-phase-locked activity. However, several lines of research suggest that these two components of oscillatory activity carry different information [46, 47]. Distinguishing them may therefore support a finer-grained dissection of computational function.

## Materials and methods

This section summarizes the methods used in the original studies [22, 23], which can be consulted for further details. The Bayesian models were first presented in [18], and that paper can be consulted for derivations of the equations.

### Subjects

34 adults (18-34 years) participated in the EEG study [23], and 30 adults (18-35 years) participated in the fMRI study [22]. Subjects had normal or corrected-to-normal vision, and no history of neurological, psychiatric, or other relevant medical problem.

### Ethics statement

All subjects provided written informed consent, which was approved by the local ethics committees (the Institutional Review Boards at Brown University and University College London).

## Experimental procedure

The experimental procedure was very similar across the two studies (see Fig 1A). Each trial began with a presentation of a visual stimulus (a colored shape in [23], a fractal in [22]) for 1000 ms. After a variable interval (250-2500 ms in [23], 250-2000 ms in [22]), a target circle appeared, at which point a response was elicited. In [23], the target appeared centrally and subjects simply decided whether or not to press a button (Go or NoGo); in [22], the target appeared laterally and subjects (if they chose to respond) indicated on which side of the screen the target appeared. After a 1000 ms delay, subjects received reward or punishment feedback. In [23], the optimal action yielded a positive outcome (reward delivery or punishment avoidance) with probability 0.7, and the suboptimal action yielded a positive outcome with probability 0.3; in [22] these probabilities were 0.8 and 0.2, respectively. Rewards were defined as monetary gains, and punishments were defined as monetary losses. Subjects were compensated based on their earnings/losses during the task.

There were 4 conditions, signaled by distinct stimuli: Go-to-Win reward, Go-to-Avoid punishment, NoGo-to-Win reward, NoGo-to-Avoid punishment. Note that subjects were not instructed about the meaning of the stimuli, so these contingencies needed to be learned from trial and error. The experimental session consisted of 40 trials for each condition in [23], 60 trials for each condition in [22], presented in a randomly intermixed order.

## EEG methods

EEG was recorded using a 128-channel EGI system, recorded continuously with hardware filters set from 0.1 to 100 Hz, a sampling rate of 250 Hz, and an online vertex reference. The EEG data were then preprocessed to interpolate bad channels, remove eyeblink contaminants, and bandpass filtered. Finally, total spectral power (both phase-locked and non-phase-locked) was computed within the theta band (4-8 Hz, 175-350 ms post-stimulus) in a midfrontal region of interest (ROI; Fig 4A) based on previous studies [48].

## fMRI methods

Data were collected using a 3-Tesla Siemens Allegra magnetic resonance scanner (Siemens, Erlangen, Germany) with echo planar imaging of a partial volume that included the striatum and the midbrain (matrix: $128 \times 128$; 40 oblique axial slices per volume angled at $-30^o$ in the antero-posterior axis; spatial resolution: $1.5 \times 1.5 \times 1.5$ mm; TR = 4100 ms; TE = 30 ms). This partial volume included the whole striatum, the substantia nigra, ventral tegmental area, the amygdala, and the ventromedial prefrontal cortex. It excluded the medial cingulate cortex, the supplementary motor areas, the superior frontal gyrus, and the middle frontal gyrus. The fMRI acquisition protocol was optimized to reduce susceptibility-induced BOLD sensitivity losses in inferior frontal and temporal lobe regions [49].

Data were preprocessed using SPM8 (Wellcome Trust Centre for Neuroimaging, UCL, London), with the following steps: realignment, unwrapping using individual fieldmaps, spatial normalization to the Montreal Neurology Institute (MNI) space, smoothing with a 6 mm full-width half maximum Gaussian kernel, temporal filtering (high-pass cutoff: 128 Hz), and whitened using a first-order autoregressive model. Finally, cue-evoked response amplitude was estimated with a general linear model (GLM), in which the event-related impulse was convolved with the canonical hemodynamic response function. The GLM also included movement regressors estimated from the realignment step.

To obtain a trial-by-trial estimate of the BOLD response at the time of the cue, we built a new GLM that included one regressor per trial at the time each cue was presented. In order to control for activity associated with the performance of the target detection task, we included a

single regressor indicating the time at which the targets were presented together with a parametric modulator indicating whether participants performed a Go (1) or a NoGo (-1) response. Similarly, to control for activity associated with the receipt of feedback, we included a single regressor indicating the time at which the outcome was presented together with a parametric modulator indicating whether the outcome was a loss (-1), a neutral outcome (0), or a win (1). Finally, the model also included movement regressor parameters. Before estimation, all regressors (except the movement regressors) were convolved with the canonical hemodynamic response function. This analysis resulted in one image per trial summarizing the BOLD response on that trial for each available voxel. We then extracted the mean BOLD response with the 2 frontal ROIs. The IFG ROI was defined as the voxels that responded to NoGo>Go in learners in the original report, thresholded at $p < 0.001$ uncorrected. The vmPFC ROI was defined as the voxels that responded positively to the parametric modulator of outcome responses in the GLM reported above, thresholded at $p < 0.001$ uncorrected.

## Computational models

We compared three computational models of learning and choice. Each model was fit to data from individual subjects using maximum likelihood estimation (see S5 Fig for histograms of the parameter estimates) and compared using random-effects Bayesian model comparison with the Bayesian information criterion approximation of the marginal likelihood [50]. We summarize the model comparison results using *protected exceedance probabilities*, which express the posterior probability that a particular model is more frequent in the population than all other models, adjusting for the probability that the differences in model fit could have arisen from the null hypothesis (uniform model frequency in the population).

Guitart-Masip and colleagues [22] compared several reinforcement learning models, finding the strongest support for one in which the action policy is defined by:

$$P(\text{Go}|s) = \frac{\exp\left[V(s, \text{Go})\right]}{\exp\left[V(s, \text{Go})\right] + \exp\left[V(s, \text{NoGo})\right]}(1 - \xi) + \frac{\xi}{2}, \tag{1}$$

where $s$ denotes the stimulus, $\xi$ is a lapse probability (capturing a baseline error rate), and $V(s, a)$ is the integrated action value for action $a$ in response to stimulus $s$:

$$V(s, \text{Go}) = V_I(s, \text{Go}) + \pi V_P(s, \text{Go}) + b \tag{2}$$

$$V(s, \text{NoGo}) = V_I(s, \text{NoGo}). \tag{3}$$

The action value integrates the instrumental value $V_I$ and the Pavlovian value $V_P$, where the weighting parameter $\pi$ captures a fixed Pavlovian approach bias towards reward-predictive cues, and an avoidance bias away from punishment predictive cues. In addition, the parameter $b$ captures a fixed Go bias. The values are updated according to an error-driven learning rule:

$$\Delta V_I(s, a) = \alpha[\rho r - V_I(s, a)] \tag{4}$$

$$\Delta V_P(s) = \alpha[\rho r - V_P(s)], \tag{5}$$

where $\alpha$ is a learning rate, $\rho > 0$ is an outcome scaling factor, and $r$ is the outcome. For the sake of brevity, we will refer to this model simply as the "RL model" (but note that the models described next could be validly considered "Bayesian RL models" insofar as they estimate expectations about reward and punishment; see [51] for further discussion of this point).

Subsequent modeling (e.g., [23]) has shown that this model can be improved by allowing differential sensitivity to rewards and punishments, but we do not pursue that extension here

since it would also require us to develop an equivalent extension of the Bayesian models described next. Since our primary goal is to model the neural dynamics underlying variability in the Pavlovian bias, we did not feel that it was necessary to run a more elaborate horse race between the model classes.

Dorfman and Gershman [18] introduced two Bayesian models. The learner is modeled as occupying one of two possible environments (controllable or uncontrollable). In the controllable environment, outcomes depend on the combination of stimulus and action, as specified by a Bernoulli parameter $\theta_{sa}$. In the uncontrollable environment, outcomes depend only on the stimulus, as specified by the parameter $\theta_s$. Because these parameters are unknown at the outset, the learner must estimate them. The Bayes-optimal estimate, assuming a Beta prior on the parameters, can be computed using an error-driven learning rule similar to the one described above, with the difference that the learning rate declines according to $\alpha = 1/\eta_s$ for the Pavlovian model, where $\eta_s$ is the number of times stimulus $s$ was encountered (the instrumental model follows the same idea, but using $\eta_{sa}$, the number of times action $a$ was taken in response to stimulus $s$). The model is parametrized by the initial value of $\eta$ and the initial values, which together define Beta distribution priors (see [18] for a complete derivation). To convert the parameter estimates (denoted $\hat{\theta}_s$ and $\hat{\theta}_{sa}$) into action values, we assumed that the instrumental values are simply the parameter estimates, $V_I(s, a) = \hat{\theta}_{sa}$, while the Pavlovian value $V_P$ is 0 for $a = \mathrm{NoGo}$ and $\hat{\theta}_s$ for Go.

The learner does not know with certainty which environment she occupies; her belief that she is in the controllable environment is specified by the probability $w$. The expected action value under environment uncertainty is then given by:

$$V(s, \mathrm{Go}) = (1 - w)V_I(s, \mathrm{Go}) + wV_P(s, \mathrm{Go}), \tag{6}$$

which is similar to the RL model integration but where the integrated action value is now a convex combination of the instrumental and Pavlovian values. Unlike the RL model, the Fixed Bayesian model used an inverse temperature parameter instead of an outcome scaling parameter (though these parameters play essentially the same role), and did not model lapse probability or Go bias (because the extra complexity introduced by these parameters was not justified based on model comparison). Thus, the action policy is given by:

$$P(\mathrm{Go}|s) = \frac{\exp\left[\beta V(s, \mathrm{Go})\right]}{\exp\left[\beta V(s, \mathrm{Go})\right] + \exp\left[\beta V(s, \mathrm{NoGo})\right]}, \tag{7}$$

where $\beta$ is the inverse temperature, which controls action stochasticity.

In the Bayesian framework, the parameter $w$ can be interpreted as a belief in the probability that the environment is uncontrollable (outcomes do not depend on actions). A critical property of the Fixed Bayesian model is that this parameter is fixed for a subject, under the assumption that the subject does not draw inferences about controllability during the experimental session. The Adaptive Bayesian model is essentially the same as the Fixed Bayesian model, but departs in one critical aspect: the Pavlovian weight parameter $w$ is updated on each trial. Using the relation $w = 1/(1 + \exp(-L))$, where $L$ is the log-odds favoring the uncontrollable environment, we can describe the update rule as follows:

$$\Delta L = r \log \frac{|\hat{\theta}_s|}{|\hat{\theta}_{sa}|} + (1 - r)\frac{1 - |\hat{\theta}_s|}{1 - |\hat{\theta}_{sa}|}. \tag{8}$$

The initial value of $L$ was set to 0 (a uniform distribution over environments).

We verified that parameters of the adaptive Bayesian model are reasonably recoverable, by simulating the experimental design used in [22], with the same number of subjects, and then fitting the simulated data. Overall, the correlation between true and recovered parameters was $r = 0.62$ ($p < 0.0001$). One parameter (the prior confidence of the Pavlovian values) exhibited relatively poor recoverability, with $r = 0.28$; we do not make any specific claims about this parameter in the paper. In addition to parameter recoverability, we found good model recoverability: the protected exceedance probability assigned to the adaptive Bayesian model was close to 1.

## Supporting information

**S1 Fig. Accuracy across conditions.** (Left) Data from Guitart-Masip et al. (2012). (Right) Simulations of the adaptive Bayesian model, using parameters fitted to the data. Error bars show standard error of the mean.
(PDF)

**S2 Fig. Pavlovian weight dynamics.** The weight variable $w$ is plotted across trial epochs, broken into quarters (note that the data sets have different numbers of trials). Error bars show standard error of the mean.
(PDF)

**S3 Fig. Disaggregated EEG results.** Midfrontal theta power (z-scored within subject) as a function of Pavlovian weight quantile, separated by stimulus condition. Error bars show standard error of the mean.
(PDF)

**S4 Fig. Disaggregated fMRI results.** BOLD response amplitude (z-scored within subject) as a function of Pavlovian weight quantile, separated by stimulus condition. Left: ventromedial prefrontal cortex. Middle: ventral striatum. Right: inferior frontal gyrus. Error bars show standard error of the mean.
(PDF)

**S5 Fig. Parameter estimate histograms.** Estimates were aggregated across the EEG and fMRI data sets.
(PDF)

## Author Contributions

**Conceptualization:** Samuel J. Gershman, Marc Guitart-Masip, James F. Cavanagh.

**Data curation:** Samuel J. Gershman, Marc Guitart-Masip, James F. Cavanagh.

**Formal analysis:** Samuel J. Gershman.

**Funding acquisition:** Samuel J. Gershman, Marc Guitart-Masip, James F. Cavanagh.

**Investigation:** Marc Guitart-Masip.

**Methodology:** Samuel J. Gershman, Marc Guitart-Masip, James F. Cavanagh.

**Project administration:** Samuel J. Gershman, Marc Guitart-Masip, James F. Cavanagh.

**Resources:** Marc Guitart-Masip, James F. Cavanagh.

**Software:** Samuel J. Gershman.

**Supervision:** Samuel J. Gershman, Marc Guitart-Masip, James F. Cavanagh.

**Validation:** Samuel J. Gershman, Marc Guitart-Masip, James F. Cavanagh.

**Visualization:** Samuel J. Gershman, Marc Guitart-Masip, James F. Cavanagh.

**Writing – original draft:** Samuel J. Gershman.

**Writing – review & editing:** Marc Guitart-Masip, James F. Cavanagh.

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
