## [Decision Letter · Decision Letter 0]

19 Aug 2020

Dear Dr. Gershman,

Thank you very much for submitting your manuscript "Neural signatures of arbitration between Pavlovian and instrumental action selection" for consideration at PLOS Computational Biology.

As with all papers reviewed by the journal, your manuscript was reviewed by members of the editorial board and by several independent reviewers. In light of the reviews (below this email), we would like to invite the resubmission of a significantly-revised version that takes into account the reviewers' comments.

Some of them are quite substantial, and will need some particular attention to make the paper suitable for publication in PLOS CB.

We cannot make any decision about publication until we have seen the revised manuscript and your response to the reviewers' comments. Your revised manuscript is also likely to be sent to reviewers for further evaluation.

Sincerely,

Daniele Marinazzo

Deputy Editor

PLOS Computational Biology

Daniele Marinazzo

Deputy Editor

PLOS Computational Biology

Reviewer's Responses to Questions

**Comments to the Authors:**

Reviewer #1: The manuscript offers a refreshing new way to look at PIT from a computational perspective. The insights that uncertainty controls arbitration between two models – a Pavlovian and a Bayesian – relies on important literature and applies it to PIT in an elegant way.

The main body of work that the authors use to motivate their model is reference [19]. They sketch two implications of the model, as follows:

• “First, the Bayesian arbitration mechanism preferentially allocates control to the Pavlovian process initially, when there are less data and hence less support for the more flexible model. This is broadly consistent with the finding that the Pavlovian bias on instrumental responding declines with the amount of instrumental training [19].” – I understand this to mean: MORE instrumental training, LESS PIT

• Second, this initial preference should be stronger in relatively less controllable environments, where little predictive power is gained by conditionalizing predictions on action. Accordingly, Pavlovian bias increases with the amount of Pavlovian training [19].” I understand this to mean: MORE Pavlovian training, MORE PIT

I may well be misunderstanding something, and therefore stand to be corrected, but reference [19] (and indeed my knowledge of the literature on which it is based) appear to say something very different: the exact set-up of the conditioning phases (prior to the PIT phase) greatly influence the pattern found; so we ignore them in our peril. The findings agree with the authors’ interpretation only for a subset of experimental set-ups, but contradict them in many others.

Reference 19 makes 3 contributions. It provides a subjective review of the literature; a meta analysis; and new empirical data.

Quoting from the review: “Greater amounts of instrumental training facilitate PIT (Holland, 2004), leading some to consider that habitual responses are more susceptible to the general motivating influence of CSs (Holland, 2004, Yin and Knowlton, 2006; see also Dickinson and Balleine, 2001). “ I understand this to mean: MORE instrumental training, MORE PIT – the opposite of the authors’ interpretation.

From the empirical work: “Extensive Pavlovian conditioning produced more Pavlovian magazine visits and weaker PIT than moderate Pavlovian conditioning (Experiment 1)”. I understand this to mean: MORE Pavlovian training, LESS PIT – again, the opposite of the authors’ interpretation.

From the meta-analysis: “The amount of instrumental training clearly influences PIT scores in non-selective transfer studies and to some extent in selective transfer studies. The precise relationship between instrumental training and PIT scores furthermore depends on the order of the instrumental and Pavlovian conditioning phases. More instrumental training facilitates PIT when Pavlovian conditioning precedes instrumental training, but appears to be detrimental to PIT when the order of the two phases is reversed.” Here, half the studies agree with the authors’ interpretation, but the other half directly contradict it.

From the meta-analysis: “There were no relationships between PIT scores and amounts of Pavlovian conditioning for groups in non-selective PIT studies. …” In “Selective PIT studies…

When Pavlovian conditioning preceded instrumental training, there was a clear negative relationship between PIT scores and the amount of Pavlovian conditioning… However, when Pavlovian conditioning followed instrumental training, there was a positive relationship between PIT scores and the amount of Pavlovian conditioning for CS-different… but not for CS-same”. My understanding here is that non-selective studies contradict the authors’ interpretation; half of the selective studies contradict it (MORE Pavlovian conditioning, LESS PIT); and some of the other half of selective studies agree with it.

For the model to be useful for the community of PIT researchers, the proposed model need to exhibit some of the agreed patterns in that literature. Of course, it’s expected that the model will make some novel predictions; but when it contradicts existing patterns, this needs to be very clearly spelled out. At present, as far as I see, the crucial reference the authors used highlights important differences between the experimental set-ups that give rise to different patterns of the relationship between training and PIT, which the current model does not have within it a mechanism to explain. It would be important, for example, to relate the key variable of interest here – controllability – to the order of training (Pavlovian vs. Instrumental first). I would love to have the author’s response to this query, because I do agree with the huge potential of their approach.

Reviewer #2: In this paper the authors reanalyze two previously published data sets (and EEG and and fMRI study) that used essentially the same valenced Go/NoGo Task that either required a Go or a NoGo response to either gain a monetary reward or avoid a monetary loss. The focus of the paper is the test of a computational model that negotiates between Pavlovian state values and instrumental state-action values using a time-varying linear combination of both value signals. The main findings from the EEG study is a dependence of frontal theta band power on the Pavlovian weight supporting the involvement of this area in the suppression of Pavlovian influence on behavior. In the fMRI data the report and interaction of behavioral response and Pavlovian weight suggesting higher activation for NoGo responses with increasing weight.

In general, the paper is well-written with a clear research question and a dedicated computational model that is being tested. Model comparison with degenerate version of the model and a previously published RL model is done with appropriate methods. I also like the idea of recycling and reanalyzing older data sets to uncover novel aspects of already established tasks. However, in the current paper, there many elements missing that one would want to see in a computational modeling study. This dampens my enthusiasm for this paper considerably at this point.

Model simulation and parameter recovery. It would be good to know that the proposed model is able to recover true parameters value during MLE estimation (Why was the estimation not done in a hierarchical manner?). In addition, it would be interesting to see, how different Pavlovian weights change the model-free signature of hte behavioral data (e.g. the interaction of response (Go/NoGo) and valence (win / neutral/ loss) that was reported in the original publications.

Model accuracy and Posterior predictive checks. We are presented with evidence that the flexible Bayesian learner fits the data best among the competing models, bug we don’t actually see, if this model is able to generate data from the fitted parameters that are commensurate of the experimental data. For instance, how accurate is the model fitting to the original experimental data and can the model replicate the finding in the original publication of an interaction of Go/NoGo and Valence?

Analysis of the fitted parameters. We never get to see the distribution of the model parameters between different participants and whether they are related behavioral performance of the imaging data.

Inconsistencies between DFs and participant number. There is an apparent inconsistency between the degrees of freedom in paired t-tests (EEG df=31, fMRI df=28) and the number participants in the two samples (EEG n=30, fMRI n=47). Were there any subjects excluded from the modeling and, if yes, for what reasons? Please resolves this inconsistency.

1st level fMRI GLM. I am confused about the 1st level GLM in the fMRI data set. There were trial-specific regressors and in addition parametric regressors for Go/NoGo and Outcome. The latter models the experimental variance due to these two factors that is common across trials, which leaves the trial-specific regressors with the residual variance that is not explained by these common factorial regressors. How is it then possible that the ROI analysis based on the trial-specific beta images is still showing effects of valence and behavioral response.

Incomplete analysis of the data. The authors correctly mention the caveat that they only analyzed responses to cue presentation, but not following the outcome leading to value updates in the model. However, they hav the data at hand to look at this question and going the extra mile would give a comprehensive picture of the neural computations associated with controllability.

Figure Permission. Figure 1b is taken directly from the original Dorfman & Gershman, NCOMMS paper, which should be cited in the Figure legend.

Reviewer #3: In this paper, the authors test a computational theory according to which Pavlovian influence will be stronger when inferred controllability of outcomes is low. They used two prior datasets of a go/no-go task in humans, and perform model-based analysis of behavior and neuroimaging data (both EEG and fMRI). They find that theta-band oscillatory power in frontal cortex tracks inferred controllability, and that these inferences predict Pavlovian action biases.

Overall, the underlying theory is very interesting and elegant but has already been published in Dorfman & Gershman (2018). The presented re-analyses of behavior, EEG and fMRI data from two previous studies are a bit superficial in the current state. A number of potential confounds have been neglected (see detailed suggestions below). Unless much more thorough analyses are presented and drastic improvements of the manuscript are made, I have the impression that not much can be learned from the paper compared to the two previous studies from which the datasets were taken, and which already showed clear neural correlates of Pavlovian influence over behavior and its modulation/suppression.

Detailed comments

The authors found an initially positive Go bias even in the Avoid condition in resp. 76% and 63% of the subjects of the datasets. This seems contradictory with the hypothesis that Pavlovian biases should intrinsically favor Go for Win and NoGo for Avoid (Huys et al., Disentangling the roles of approach, activation and valence in instrumental and pavlovian responding. PLoS Comput Biol 7, e1002028 (2011).). How do the authors reconcile their observation with the underlying theoretical hypothesis?

The tested model space is too small. Why not comparing to a non-Bayesian RL model with annealed learning rate, similar to the Bayesian model, so as to assess the specific role of this annealing process? It would also be interesting to compare the model to a fixed amplitude change model rather than based on RPE amplitude (to assess the variability of magnitude changes), and to compare the model to a random walk model (to assess the consistency of direction changes).

The authors systematically plot variables of interest against weight quantile (e.g., Figs. 2, 3). This is interesting but not sufficient to evaluate the temporal evolution of these variables. In Dorfman & Gershman (2018; Fig. 6), the Go bias decreases rapidly in less than 10 trials, and then remains flat during the rest of the experiment. Is this also the case here? Is it also the case for the Pavlovian weight and for the variables of interest plotted in Figs. 2 and 3? Figure S1 shows the evolution of the Pavlovian weight through time. However, it is important to plot the trial-by-trial evolution, and not just for different quarters of trials. Moreover, it is important to show distinct plots for different conditions and different groups of subjects (see next paragraph). Finally, because the Go bias and Pavlovian weights are expected to change a lot during early experiment and then to remain nearly flat, it is important to verify that correlations with the model’s Pavlovian weight still hold when only considering the second half of each condition block.

In Cavanagh et al. (2013), there are important performance difference between the four conditions as well as between learners and non-learners. Are there differences in model fitting accuracy and in model parameters between conditions or between learners and non-learners? How does the evolution of Pavlovian weight with time differs between these cases? And how does the correlation between frontal theta and Pavlovian weight differ between these cases?

From the methods, it is not clear that the four conditions are presented in distinct blocks of trials. In contrast, this is clear from the original papers. I think this should be specified here too. Moreover, I think the results would be different if trials from the four conditions were intermixed, and this should be discussed here. Finally and more importantly, it is not clear to me whether the order between conditions was counterbalanced between subjects or not. This is important since some prior knowledge can be used by subjects and learning can be facilitated during late blocks (especially the fourth one) based on the previously encountered task rules. For instance, in the data of Guitart-Masip et al. (2011), how can one disentangle this effect from the Pavlovian effect to explain the better performance in the fourth block (NoGo to Avoid) than in the third block (NoGo to Win)? Were there significant differences in the initial values of the fitted model between conditions in any of the two datasets? And how did this affect learning and the evolution of the Pavlovian bias?

The authors verified that other model variables, like instrumental and Pavlovian values, do not correlate with the Pavlovian weight, and thus are not potential confounds. But other potential confounds should also be tested here, like choice confidence, reward uncertainty and reward prediction errors, to make sure that the frontal theta power is here only reflecting the suppression of the Pavlovian influence on choice. Along these lines, it has been shown previously that midfrontal theta may relate to reward prediction errors (Holroyd CB, Krigolson OE, Lee S (2011) Reward positivity elicited by predictive cues. Neuroreport 22:249 –252), to behavioral slowing (Cavanagh JF, Frank MJ, Klein TJ, Allen JJ (2010) Frontal theta links prediction errors to behavioral adaptation in reinforcement learning. Neuroimage 49:3198 –3209), and switching (Cohen MX, Ranganath C (2007) Reinforcement learning signals predict future decisions. J Neurosci 27:371–378; van de Vijver I, Ridderinkhof KR, Cohen MX (2011) Frontal oscillatory dynamics predict feedback learning and action adjustment. J Cogn Neurosci 23:4106–4121). Could the authors here control for these potential predictors of variations in frontal theta power, and whether the Pavlovian bias in the model can account for these effects?

About the EEG data, it is a bit unsatisfying to just show a basic correlation between frontal theta power and Pavlovian bias. Canavagh et al. (2013) found that ‘interindividual differences in theta power increases in response to Pavlovian conflict (across participants) correlated with intraindividual abilities to use theta (across trials) to overcome Pavlovian biases’. Could the authors here also assess the role of Pavlovian conflict and interindividual differences in the modulation of the correlation between frontal theta power and the Pavlovian bias?

The basic analyses of fMRI data presented here are not satisfying either. For instance, it has been shown that vMPFC encodes option values and confidence (M. Lebreton, R. Abitbol, J. Daunizeau, M. Pessiglione (2015) Automatic integration of confidence in the brain valuation signal, Nat. Neurosci., 18(8):1159‑1167; B. De Martino, S. Bobadilla-Suarez, T. Nouguchi, T. Sharot, B. C. Love (2017) Social Information Is Integrated into Value and Confidence Judgments According to Its Reliability, J. Neurosci., 37(25):6066‑6074). Could the authors check for these potential confounds of vMPFC activity, draw links with the model?

Some methodological information is missing. For instance, is the reported frontal theta power phase-locked or not, and what does this imply in terms of interpretation? In the fMRI data, to which events of the task are the IFG and vMPFC responses shown?

Typos

Page 6, differed fom 0 -> from 0.

Pages 7 and 11, please fix three occurrences of ‘NoGo¿Go’.

**Have all data underlying the figures and results presented in the manuscript been provided?**

Reviewer #1: None

Reviewer #2: Yes

Reviewer #3: Yes

PLOS authors have the option to publish the peer review history of their article (what does this mean?). If published, this will include your full peer review and any attached files.

Reviewer #1: No

Reviewer #2: No

Reviewer #3: No
---

## [Decision Letter · Decision Letter 1]

16 Oct 2020

Dear Dr. Gershman,

Thank you very much for submitting your manuscript "Neural signatures of arbitration between Pavlovian and instrumental action selection" for consideration at PLOS Computational Biology.

This version is much improved. Still some important issues have not been addressed. We appreciate that you may disagree with these, still it would be good to discuss, both before and after an eventual publication. . In light of the reviews (below this email), we would like to invite the resubmission of a significantly-revised version that takes into account the reviewers' comments.

We cannot make any decision about publication until we have seen the revised manuscript and your response to the reviewers' comments. Your revised manuscript is also likely to be sent to reviewers for further evaluation.

Sincerely,

Daniele Marinazzo

Deputy Editor

PLOS Computational Biology

Daniele Marinazzo

Deputy Editor

PLOS Computational Biology

Reviewer's Responses to Questions

**Comments to the Authors: **

Reviewer #1: I really like the author’s modelling of the neural data, and agree that their project makes sense as a follow up and consolidation of the earlier modelling of behavioural data. The results of the analysis of midfrontal theta and VMPFC effects are novel and would be useful to others who work in this field.

I regret that in my view, the authors’ response to my critique was not sufficiently robust. Put simply, the original manuscript was motivated by behavioural findings X and Y; therefore, there need to be a bit more reconning, when it is pointed out that the findings are actually ~X and ~Y. 

While I agree with the authors that the go-no-go task could, in principle, rely on different principles than the PIT task, and it is absolutely right to point out any links between these tasks should be tested empirically, the logic of the model is based on the hypothesis that the task taps generic Pavlovian and Instrumental processes. This is, indeed, how the abstract and introduction are written. 

Therefore, when they say that the “Pavlovian weight can also be interpreted as the subjective degree of belief in an uncontrollable environment”, it makes sense for readers to reflect on situations where an agent is trained less well in the instrumental task, and therefore, their belief in the controllability of the environment should be lower. When the same agent is given an opportunity to behave in a way that can reflect both instrumental and Pavlovian biases, then if I had understood the model correctly, the Pavlovian biases should be stronger when instrumental training was extensive, and weaker when instrumental training was less extensive. Yet in this situation, during a PIT test, the empirical finding is exactly the opposite. As the PIT situation is the best-studied example of the interaction of Pavlovian and instrumental processes, readers are likely to have this example in mind.

The work and the data remain interesting if the model can only explain the go-no-go task; or tasks where the stimuli are spaced in particular ways; but this needs to be stated clearly in the introduction with the boundaries laid out.

Minor comments

1.“At the time of stimulus presentation, the reward prediction error is simply the stimulus value – “ Would participants not update the reward prediction based on their performance on previous trials, and the reward offered on previous trials, such that the reward they can predict at the start of the new trial varies somewhat? 

2.In a number of places, the authors rule out alternative interpretation of the data by referring to null results of a signed rank test – e.g. three places on p. 6 and one place at the top of p. 7. This is a worry, because while the sample may have been powered for the purpose of the original study, it may not have been sufficiently large to allow the detection of the effects the authors refer to here (e.g. if it was powered to detect difference between means, it may not be large enough to detect a significant correlation). Please could the authors state the statistical power they had in every case they refer to a null result, e.g. “the median correlation never significantly differed from 0, although we only had power to detect effects that are medium in magnitude, or higher”.

Reviewer #2: The authors have addressed my all my questions sufficiently. I support the publication of this paper now.

**Have all data underlying the figures and results presented in the manuscript been provided?**

Reviewer #1: None

Reviewer #2: Yes

PLOS authors have the option to publish the peer review history of their article (what does this mean?). If published, this will include your full peer review and any attached files.

Reviewer #1: No

Reviewer #2: No
---

## [Decision Letter · Decision Letter 2]

23 Nov 2020

Dear Dr. Gershman,

We are pleased to inform you that your manuscript 'Neural signatures of arbitration between Pavlovian and instrumental action selection' has been provisionally accepted for publication in PLOS Computational Biology.

Best regards,

Daniele Marinazzo

Deputy Editor

PLOS Computational Biology

Daniele Marinazzo

Deputy Editor

PLOS Computational Biology

Reviewer's Responses to Questions

**Comments to the Authors:**

Reviewer #1: Although my own preference would be for another sentence in the discussion where the authors refer to PIT - perhaps pointing out that there are potential discrepancies to resolve in future, I agree that the revised manuscript presents a fair, and far more cautious, interpretation of the results. The revised manuscript is far improved in my view, and I recommend that it is now published. I thank the authors for addressing my reservations.

**Have all data underlying the figures and results presented in the manuscript been provided?**

Reviewer #1: Yes

PLOS authors have the option to publish the peer review history of their article (what does this mean?). If published, this will include your full peer review and any attached files.

Reviewer #1: No

---

## [Editor Report · Acceptance letter]

2 Feb 2021

PCOMPBIOL-D-20-00957R2 

Neural signatures of arbitration between Pavlovian and instrumental action selection

Dear Dr Gershman,

I am pleased to inform you that your manuscript has been formally accepted for publication in PLOS Computational Biology. Your manuscript is now with our production department and you will be notified of the publication date in due course.

With kind regards,

Alice Ellingham
